# The Effect of Tropical Cyclone Nicholas (11–20 February 2008) on Sea Level Anomalies in Indonesian Waters

**Nining Sari Ningsih [1,\*], Farrah Hanifah [1], Tika Sekar Tanjung [2], Laela Fitri Yani [2] and Muchamad Al Azhar [3]**

[1] Oceanography Research Group, Faculty of Earth Sciences and Technology, ITB, Bandung, Jawa Barat 40132, Indonesia; farrah@oceanography.itb.ac.id

[2] Study Program of Oceanography, Faculty of Earth Sciences and Technology, ITB, Bandung, Jawa Barat 40132, Indonesia; tika.sekar15@gmail.com (T.S.T.); laelafitri.pri@gmail.com (L.F.Y.)

[3] Plymouth Marine Laboratory, Plymouth PL1 3DH, UK; maz@pml.ac.uk

\* Correspondence: nining@fitb.itb.ac.id

**Abstract:** As reported extensively in both electronic and print media in Indonesia, high wave and anomalously high sea level phenomena occurred in February 2008 in Indonesian waters, mainly along the western coast of Sumatra and the southern coasts of Java-Bali. Tropical Cyclone (TC) Nicholas, occurring in northwestern coastal waters of Australia between 11 and 20 February 2008, might have contributed to the existence of these phenomena in the Indonesian region. This study focused on investigating the effect of TC Nicholas on the increases in sea levels in the Indonesian waters by analyzing residual water levels (non-astronomic tide). In this regard, a storm tide event (the sum of the astronomical tide and storm surge generated by the TC Nicholas) was simulated in this region using the Regional Ocean Modeling System (ROMS). The residual water levels were obtained by removing the tidal part (astronomic tide) from the ROMS simulated total water levels. In addition, to confirm possible influences of TC Nicholas, a lagged correlation analysis was applied between atmospheric pressure at the center of TC Nicholas and residual water level oscillations in the Indonesian waters. It was found that the residual water levels showed a strong correlation with the atmospheric pressure at the center of TC Nicholas in some areas of the Indonesian seas, such as the western coast of Sumatra, the southern coast of Java, Lesser Sunda Islands, and the southern coast of Papua. The increased sea levels on the western coast of Sumatra are up to 16 cm, with TC Nicholas leading the residual water level by 4.18 days (TL: time lag). Meanwhile, they are up to 20 cm (TL = 5.75 days), 21 cm (TL = 1.12 days), and 38 cm (TL = 3.96 days) on the southern coast of Java, the Lesser Sunda Islands, and the southern coast of Papua, respectively. The results of this study could be used as an initial assessment to investigate the most vulnerable Indonesian coastal areas to the impact of the TC and they might be significantly beneficial for designing both a proper disaster risk reduction program and investment policies in the region, particularly in the context of flood risk reduction and adaptation.

**Keywords:** Tropical Cyclone Nicholas; sea level anomalies; increases in sea levels; residual water levels; ROMS; Indonesian waters

## 1. Introduction

Indonesia is an archipelago located on the equator with a coastline of around 108,000 km, making it the second-longest coastline in the world [1]. This large expanse of coastal area has the

potential for significant transportation, industrial, residential, port, and even recreational activities. However, the coastal areas are also vulnerable to marine disasters, such as abrasion, tsunamis, and storm tides. Several studies have been carried out in the Indonesian waters and its surrounding sea areas, especially in the Indian Ocean's eastern boundary or Southeastern Tropical Indian Ocean (SETIO). The SETIO adjacent to the southern coasts of Sumatra and Java plays an important role in climatic and oceanic variability and complex dynamical processes exist in this region, such as the monsoon system [2–4], South Java Current (SJC) [5,6], South Java Undercurrent (SJUC) [7], South Equatorial Current (SEC) [8,9], Indonesian Throughflow (ITF) flowing from the major exit passages (e.g., Sunda Strait and the Lesser Sunda Island chain: Lombok and Ombai Straits, and Timor Passage) [10,11], Indian Ocean Dipole (IOD) [12], eddies [13–15], Rossby waves [16,17], and Kelvin waves [18–20]. Moreover, investigation of these dynamical processes in the SETIO region has been of particular interest for both scientific and practical reasons, such as understanding upwelling variability and mixing for fisheries [21–23] and investigating ocean wave energy as a potential renewable energy resource [24,25].

In addition to the important role of the SETIO in ocean and atmosphere variability, this region, especially close to the western coast of Sumatra and the southern coasts of Java and Lesser Sunda Islands, is not only prone to impact of high waves, such as tsunami [26,27], wind sea, and swell generated by tropical cyclones in the Indian Ocean [28,29], but also to sea level anomalies, especially increases in sea levels due to storm tide (the combination of storm surge and the astronomical tide) [30], storm surge (the transient changes due to the effects of non-astronomic tide, such a tropical cyclone or storm), and long-term changes (e.g., sea level rise due to climate change) [31,32]. Sea level anomalies caused by storm surge and their implication have always challenged scientists. This subject has been intensively studied, e.g., remote forcing contribution to anomalously high sea levels along the Florida coast of Apalachee Bay during Hurricane Dennis [33], the role of tide-surge interaction in the distribution of surge residuals in the North Sea [34], the identification of storm surge vulnerable areas along the east coast of India [35], the simulation of storm surge on the Bangladesh coast [36], the estimation of extreme water levels resulting from astronomical tides and surge residuals in Irish coastal waters [37], and the simulation of tides and hurricane-induced storm surges in the Gulf of Mexico [38]. Although extensive studies of this subject have been carried out in the aforementioned regions, this kind of research in the SETIO adjacent to the Indonesian seas is still limited. Most importantly, the SETIO is the part of the Australian tropical cyclone region where tropical cyclones develop in the Southern Hemisphere [39]. It is important to determine and monitor the increased sea levels due to tropical cyclones, especially related to flood defence design and management. However, this important subject has so far not been extensively studied in the Indonesian seas and its coastal areas. In this regard, an investigation of increases in sea levels due to a tropical cyclone in this region will be of particular interest.

Indonesia's geographical position, which is between 6° N–11° S and 95° E–141° E, is not in the path of tropical cyclones (TCs). However, the presence of tropical cyclones in proximity to Indonesia, especially those formed around the northwest Pacific and southeast Indian Oceans, as well as Australia, do influence weather patterns in Indonesia. Changes in weather patterns as the result of TCs indicate that tropical cyclones have an indirect impact on weather conditions in Indonesia, such as strong winds and high swell waves [40]. TCs that have occurred in northwest Australia, such as Jacob and George (2–12 March 2007), Nicholas (11–20 February 2008), and Marcus (14–27 March 2018), and those that have occurred relatively close to the Indonesia region, such as Cempaka and Dahlia (22 November to 4 December 2017), have contributed to both the generation of large waves and increases in water levels in the Indonesian seas. A study conducted by Ningsih et al. [30] shows an increase in elevation of water level due to storm surges of about 19 cm at Nusakambangan (southern coast of West Java) when the TCs Jacob and George occurred in March 2007, as well as approximately 9–13.5 cm at several other points on the southern coast of Java. Moreover, as reported by an Indonesian print media [41], sea wave heights at Yogyakarta (southern coast of Central Java) and Bali reached 5 m, those in East Nusa Tenggara waters reached 6 m, and those on the western coast of Sumatra reached 4 m as a result of TC Nicholas (February 2008) [41]. In recent years, when

TC Marcus occurred in March 2018, the Indonesian National Agency for Disaster Countermeasure or Badan Nasional Penanggulangan Bencana (abbreviated as BNPB) stated that it impacted the height of waves, rising to 2.5 to 4 m. This took place in several areas, such as the southern Sunda Straits, as well as the southern coasts of Java and Bali. Meanwhile, during the TCs Cempaka and Dahlia, which occurred relatively close to the Indonesia region, the simulation results of a numerical wave model carried out by Windupranata et al. [28] showed that the highest wave height of 3.75 m took place at Ujung Genteng (the southern coast of West Java).

Ramdhani [29] has investigated the characteristics of TCs developed in the Australian tropical cyclone region that will generate high waves in the Indonesian waters using 24 years' (January 1988–December 2011) data derived from simulated results of the WAVEWATCH-III (WW3) wave model. There were 165 TCs that occurred in the region during the 24-year simulation. The study of Ramdhani [29] has revealed several conditions that are needed for a tropical cyclone in the Australian tropical cyclone region to generate high waves in Indonesian waters, namely, (1) TC intensity is greater than or equal to tropical storm (TS); (2) the period of occurrence of the TC is December–February (DJF); and (3) the TC moves towards the western coast of Australia (TC track). Additionally, it has been revealed that the TCs occurring during DJF will strengthen northwest wind in the Indonesian region [29]. The simulated results of WW3 showed that TC Nicholas, occurring in February 2008 and moving towards the western coast of Australia, contributed to the existence of large waves in the Indonesian waters, although its intensity was relatively weak (Category 1). On the other hand, TC Inigo that occurred during 1–8 April 2003 had not caused high waves in the Indonesian waters in spite of the fact that its intensity (Category 4) was stronger than that of TC Nicholas (Category 1). It has been suggested that the existence of large waves in the Indonesian waters is not only determined by the TC intensity, but also the TC occurrence period and the TC track [29].

Almost all the aforementioned studies and reports deal with the effect of TCs on the generation of large waves (wind sea and swell) with wave periods less than 20 s. However, detailed characteristics of the TCs impacts on increases in sea levels, especially dealing with surge residuals (non-astronomic tide), are still limited and have not been fully explained in the Indonesian waters. This is the main motivation of the present study. In this present study, we focused on the TC Nicholas event (February 2008) for this analysis, because it met the aforementioned criteria for generating large waves in the Indonesian waters [29]. In addition, the occurrence of big waves in the Indonesian seas in February 2008 was reported extensively in both Indonesian electronic and print media. During the occurrence of TC Nicholas, the Indonesian Agency for Meteorology, Climatology, and Geophysics (Badan Meteorologi, Klimatologi, dan Geofisika/BMKG) warned that the sea conditions are dangerous for fishermen and sailors to be out at sea. Moreover, ships carrying wood and iron at the Gresik Port also had to postpone their voyages because of the high waves, as BMKG banned ships from sailing for three weeks in February 2008 [42].

Therefore, it is necessary to acquire better and comprehensive insights of the impact of TC Nicholas on increased sea levels, as well as the occurrence of large waves in the Indonesian waters. To the best of our knowledge, this important subject has so far not been extensively investigated in the region. Hence, in this present study, we aim to investigate increased sea levels due to the TC Nicholas event in the Indonesian waters by analyzing residual water levels (non-astronomic tide). The residual water levels were acquired by removing the tidal component (astronomical tide) from the total water levels, which are derived from simulated results of a hydrodynamic model known as the Regional Ocean Modeling System.

## 2. Materials and Methods

In this study, the increased sea levels due to the TC Nicholas were quantified by examining the residual water levels (total water level minus the astronomic tide). The total water levels were obtained from the simulated results of the Regional Ocean Modeling System (ROMS), which was developed by Shchepetkin and McWilliams [43]. In this regard, T_TIDE software [44], a package of routines that can be utilized to conduct classical harmonic analysis, was used to separate astronomical and non-astronomical components of the ROMS simulated total water levels. Here, we considered

the residual sea levels (non-astronomical component), which are only influenced by storm surge, inverted barometer effect (hereafter refer to IBE), and wind set-up (known as meteorological induced sea level rise). Therefore, we neglected the influences of the long-term changes (e.g., sea level rises due to climate change). Furthermore, to estimate the increased sea levels in the Indonesian coastal areas due to remote forcings during the TC Nicholas event, influences of local forcings (wind set-up and IBE), which are calculated by Equations (4)–(6), were subtracted from the residual sea levels. In this case, remote forcings are referred to as the difference between the residual sea levels at observed sites and the locally calculated IBE and wind set-up. Additionally, a lagged correlation analysis [45] was implemented between atmospheric pressures at the center of TC Nicholas and residual sea level oscillations in the Indonesian waters to corroborate possible effects of the cyclone. A power spectral analysis [45] was then applied to the residual sea levels at the observed sites and atmospheric pressures at the center of TC Nicholas to identify dominant periods of their variations.

## 2.1. Model Description and Its Application

The ROMS used in this study was widely applied to simulate hydrodynamics of estuarine, coastal, and oceanic waters both for scientific and practical purposes (e.g., [46–51]). The governing equations solved in the ROMS model are the primitive equations for momentum, mass, heat, and salt. In the horizonal, the primitive equations are formulated using boundary-fitted, orthogonal curvilinear coordinates on a staggered Arakawa C-grid. Meanwhile, stretched terrain-following coordinates are used in the vertical to discretize the governing equations. The horizontal curvilinear system allows to resolve a complex geometry of the coastline domain, whereas the stretched terrain-following coordinates enable to increase the resolution in research areas of interest, such as the surface and bottom mixed layers. In addition, the model comprises a mode splitting technique, solving the horizontal two-dimensional mode (barotropic/external mode) for the fast processes and the three-dimensional mode (the baroclinic/internal mode) for slower processes. Further details of the ROMS can be obtained in Shchepetkin and McWilliams [43].

Figure 1 shows the study areas used to simulate the storm tide event, which was generated by the astronomical tide and TC Nicholas. The model domain covers all of the Indonesian waters, South China Sea, southwestern Pacific Ocean, and northern parts of Australia. The resolution of the model domain is 1/8° or equivalent to 13.85 km, which consists of 500 × 388 grids. The model was run for 60 days (2 January to 29 February 2008), which covered periods of the TC Nicholas occurrence (11–20 February 2008). The first 30 days were for the spin-up, whereas the subsequent 30 days were used for analysis. The storm tide event was simulated by imposing tidal elevations at the open boundaries, winds, atmospheric pressures, as well as TC Nicholas intensity (wind field and atmospheric pressure) from the observed track parameters. Moreover, the TC Nicholas track is also included in Figure 1.

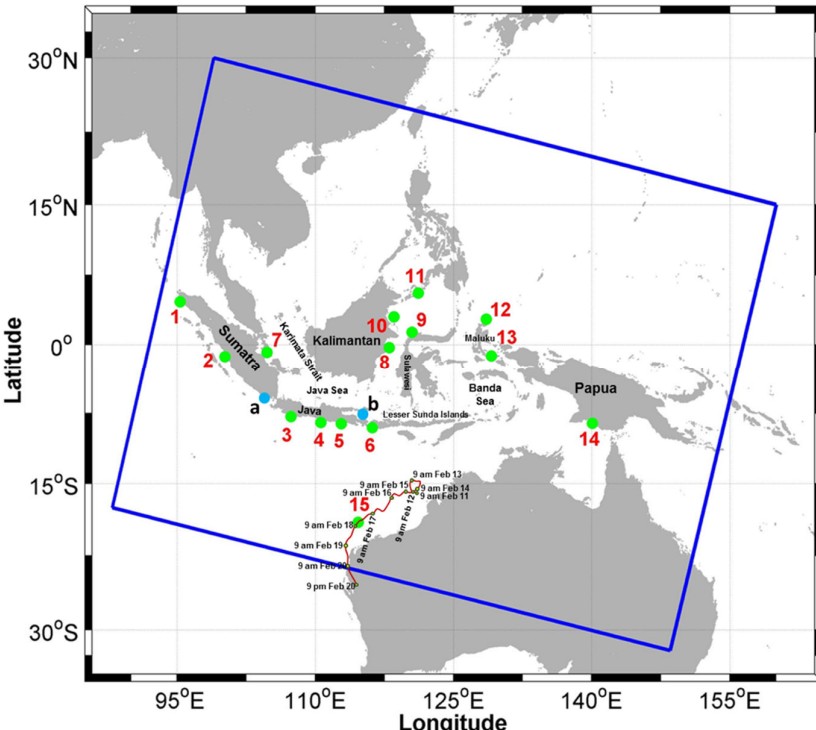

**Figure 1.** Model domain (88° E–162.38° E and 31.59° S–29° N). Numbers (1–15) indicate locations of verification points for astronomic tide (names of the verification points shown in Table 1). Alphabet letters a (Lampung) and b (Buleleng) denote locations of the validation point for meteorological induced sea level variations (non-astronomical tides). Red line indicates the track of Tropical Cyclone (TC) Nicholas during 11–20 February 2008.

*2.2. Data*

Bathymetric data used in this research were derived from Earth Topography with two minutes gridded (ETOPO 2) (https://www.ngdc.noaa.gov/mgg/global/etopo2.html) combined with digitized bathymetric maps at 1:20,000 scale and at an average sounding spacing of about 0.73 km (0.41 min), which are issued by the Hydrography and Oceanography Center of the Indonesian Navy (HOCIN). The HOCIN map was used to improve the quality of the bathymetric data in Makassar Strait, Java Sea, and Jakarta Bay. This is implemented because the ETOPO2 data have a low level of accuracy for waters with depths less than 200 m according to Sindhu et al. [52]. Meanwhile, tidal data imposed at the open boundaries were obtained from Oregon State University's TOPEX/Poseidon Global Inverse Solution Tidal Model Version 7.2 (TPXO 7.2) (http://volkov.oce.orst.edu/tides/TPXO7.2.html) based on 10 tidal constituents ($M_2$, $S_2$, $N_2$, $K_2$, $K_1$, $O_1$, $P_1$, $Q_1$, $M_f$, dan $M_m$).

Surface winds, surface specific humidity, mean sea level pressure, surface air temperatures, downward long and short wave radiations at the surface, precipitation, and evaporation data were used to provide atmospheric forcing for ROMS. These atmospheric data were taken from the European Centre for Medium-Range Weather (ECMWF) Re-Analysis-Interim (ERA-Interim) with a time interval of three hours and a grid resolution of 1/8° (https://apps.ecmwf.int/datasets/data/interim-full-daily/levtype=sfc). In this study, because the intensity of TC Nicholas (wind field and atmospheric pressure data) taken from the ERA-Interim at this spatial/temporal resolution will not capture the TC intensity correctly, we blended these ERA-Interim ambient data with the observed track parameters of TC Nicholas (e.g., positions, wind fields, central atmospheric pressure, and radius), which were provided by the Australian Bureau of Meteorology (BOM) (http://www.bom.gov.au). TC Nicholas, occurring from 11–20 February 2008, was a Category 1 cyclone according to the Saffir-Simpson Hurricane Scale, with the lowest atmospheric pressure of 948 hPa and maximum wind speed of about 80 knots (41.15 m/s). According

to the BOM, TC Nicholas was a long-lived cyclone persisting for 10 days, which unusually moved slowly towards the western coast of Australia at on average 4–10 km/h until 16 February 2008 and increasing to 10–13 km/h in the subsequent stages.

Because of the lack of tide gauge data in the Indonesian waters during the occurrence of TC Nicholas, the ROMS simulated total water levels could not be verified directly against observed data. In this regard, we only verified astronomic and non-astronomic components of the ROMS simulated total water levels. The astronomic component of the ROMS results (the simulated astronomic tide) was validated against tidal prediction data provided by the Indonesian Geospatial Information Agency (Badan Informasi Geospasial, BIG) at 15 points, as shown in Figure 1. Meanwhile, the non-astronomic component of the ROMS results (the simulated non-astronomic tide or the meteorological induced sea level variation) was only compared to weekly data of TOPEX/Poseidon (Ocean Topography Experiment) Sea Level Anomaly (SLA) with a spatial resolution of 1/3° (http://apdrc.soest. hawaii.edu). As an example, in this paper, we only showed the validation of non-astronomical tide at two points, namely Lampung and Buleleng, marked by alphabet letters a and b, respectively, in Figure 1.

### 2.3. Verification Method

In this study, the correlation coefficient, normalized root mean square error (NRMSE), and bias are used to verify the simulated astronomic tide at Points 1–15 in Figure 1 using the following formulation:

$$\text{Correlation coefficient } (R) = \frac{\sum_{i=1}^{n}(x_i-\bar{x})(y_i-\bar{y})}{\sqrt{\sum_{i=1}^{n}(x_i-\bar{x})^2 \sum_{i=1}^{n}(y_i-\bar{y})^2}}, \tag{1}$$

$$\text{NRMSE} = \frac{\sqrt{\frac{1}{n}\sum_{i=1}^{n}(y_i-x_i)^2}}{y_{max}-y_{min}} \text{ x } 100\%, \tag{2}$$

$$\text{Bias} = \bar{y} - \bar{x}, \tag{3}$$

where $x_i$ is the simulated astronomic tide (m); $y_i$ is the BIG tidal prediction (m); $\bar{x}$ is temporal mean simulated astronomic tide (m); $\bar{y}$ is temporal mean BIG tidal prediction (m); $y_{max}$ and $y_{min}$ are maximum and minimum values of the BIG tidal prediction (m), respectively; $i$ is sequence data; and $n$ is the amount of data. In this regard, the NRMSE is RMS error with respect to the BIG tidal data range.

### 2.4. Local Forcing Effects

As mentioned before, changes in sea level due to wind set-up and IBE were calculated locally at the observation sites (Points 1–16 in Table 2 and Figure 6) to assess the influences of local forcings on the increases in sea levels during the TC Nicholas event. Furthermore, the influences of remote forcing during the TC occurrence are estimated by subtracting the calculated local forcings (wind set-up and IBE) from the meteorological induced sea level variation (the simulated non-astronomic tide or the residual sea levels). This kind of approach of local and remote forcing influences has also been applied by Ningsih et al. [53] to study storm surge characteristics (e.g., surge-heights and periods) along the northern coasts of Java based on tide gauge data and TC events occurring in the South China Sea.

Changes in elevation caused by wind friction (wind stress) on the surface of the water are called wind set-up. This process can occur when winds move towards a barrier, such as a beach. Changes in sea level due to wind set-up can be calculated using the following formulation:

$$\Delta\eta = C\frac{\tau_{sx}}{\rho g h}\Delta x, \tag{4}$$

$$\tau_{sx} = \rho_{air}C_D W_x|W_x|, \tag{5}$$

where $\Delta\eta$ denotes changes in sea level due to wind set-up over a horizontal distance $\Delta x$ perpendicular to shoreline (m); $C$ is coefficient in which $1 < C < 1.5$; $\tau_{sx}$ is wind stress in the normal direction to the shore (N/m²); $\rho$ is the density of sea water (kg/m³); $g$ is the earth gravity acceleration (m/s²); $h$ is water depth (m); $\rho_{air}$ is the air density (kg/m³); $C_D$ is the wind drag coefficient; and $W_x$ is the wind speed perpendicular to shoreline (m/s). Meanwhile, changes in elevation due to changes in atmospheric pressure $\Delta\eta_p$, which are also referred to as the inverted barometer effect (IBE), are formulated as follows:

$$\Delta\eta_P = -\frac{1}{\rho g}\Delta P_a, \tag{6}$$

where $\rho$ is sea water density (kg/m³) and $\Delta P_a$ is changes in atmospheric pressure (N/m²). Theoretically, the formulation can be interpreted to indicate that any increase in atmospheric pressure will be followed by a reduction in sea level. If atmospheric pressure is expressed in units of mb (millibars), then a reduction of 1 mb ($10^3$ mb = $10^5$ N/m²) in pressure will be followed by an increase in sea level of 1 cm, and vice versa.

## 3. Results and Discussions

Firstly, we described the performance of the model results by comparing the simulated astronomic tide with the BIG tidal prediction and the simulated non-astronomic tide (the meteorological induced sea level variation) with the weekly TOPEX/Poseidon SLA. As already explained, the simulated total water levels could not be verified directly against observed data because of the lack of measurements of sea levels from tide gauges in the Indonesian coastal areas during the TC Nicholas event. Secondly, the areas most affected by the TC Nicholas event will be analyzed by assessing maximum values of the increased sea levels as well as the lag between the peak of TC Nicholas activity and the response in sea levels based on the simulated non-astronomic tide (the residual sea levels).

### 3.1. Model Verification

Table 1 shows detailed values of the normalized root mean square error (NRMSE), bias, and phase lag between the simulated astronomic tides and those of the BIG at the verification sites (Points 1–15 in Figure 1). The ranges of the NRMSE and bias between the simulated values of astronomic tides and those of the BIG are about 5–20% and about −0.5 and 0.25 cm, respectively. Meanwhile, in general, the phase lags between them are less than 2 h (Table 1). Figure 2 shows an example of verification for the astronomic tides at Tarakan (marked by Point 10 in Figure 1). The scatter diagram for the simulated astronomic tides (model) and the BIG tidal prediction (BIG) shows clustering of points approximately around the line of the model equal to the BIG (Figure 2b). The general agreement between the simulated results and those of the BIG is reasonably encouraging (Table 1 and Figure 2).

**Table 1.** Values of the normalized root mean square error (NRMSE), biases, and phase lags between the simulated astronomic tides and those of the Badan Informasi Geospasial (BIG).

| Points | Locations | NRMSE (%) | Bias (cm) | Phase Lags (hour) |
|--------|-----------|-----------|-----------|-------------------|
| 1 | Aceh | 15 | −0.30 | −1 |
| 2 | Padang | 12 | 0.01 | −1 |
| 3 | Garut | 15 | 0.12 | −1 |
| 4 | Yogyakarta | 16 | 0.03 | −1 |
| 5 | Malang | 16 | 0.01 | −1 |
| 6 | Koeta, Lombok | 16 | −0.04 | −1 |
| 7 | Jambi | 09 | −0.17 | 1 |
| 8 | Samarinda | 20 | −0.19 | 2 |
| 9 | Tolitoli | 11 | −0.20 | 1 |
| 10 | Tarakan | 11 | −0.17 | 1 |
| 11 | Tipo-Tipo, Filipina | 11 | −0.19 | 1 |

| 12 | Hapo, Maluku | 05 | −0.25 | 0 |
| 13 | Obi, Halmahera | 18 | −0.24 | 1 |
| 14 | Merauke | 11 | 0.25 | 1 |
| 15 | Western Australia | 17 | −0.50 | −1 |

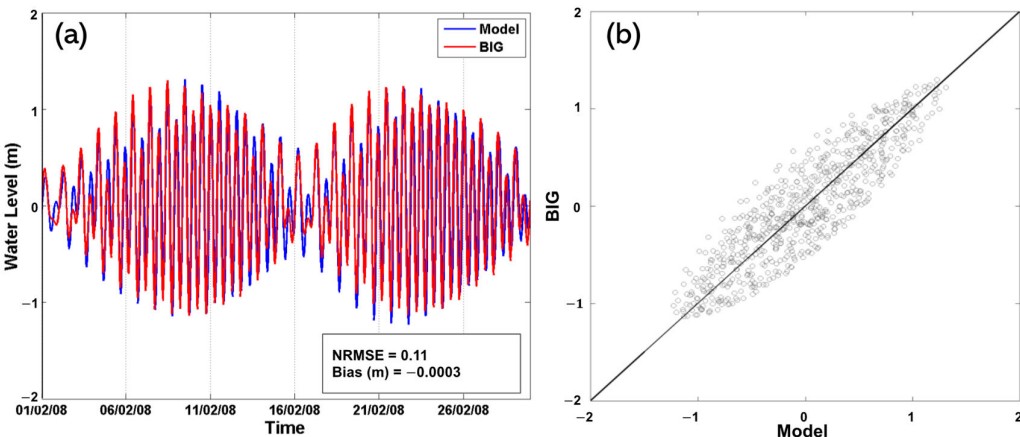

**Figure 2.** Validation of the astronomical sea level between the model results and the Badan Informasi Geospasial (BIG) predicted data at the Tarakan (marked by Point 10 in Figure 1): (**a**) time series; (**b**) scatter diagram. NRMSE, normalized root mean square error.

Furthermore, to know the model performance in simulating meteorological induced sea level variations, we performed comparisons between the non-astronomic component of the ROMS results and the weekly TOPEX/Poseidon SLA. Figure 3 shows examples of the validation of non-astronomical tide at Lampung and Buleleng, marked by the letters a and b in Figure 1, respectively. Additionally, because we are interested in assessing maximum values of the increased sea levels induced by meteorological effects, comparison of the maximum values of the non-astronomic tide between the simulated results and those of the TOPEX/Poseidon (Figure 4) were carried out at 16 locations (denoted by Points 1–16 in Table 2 and Figure 6), which were most susceptible to the impact of the storm surge generated by TC Nicholas.

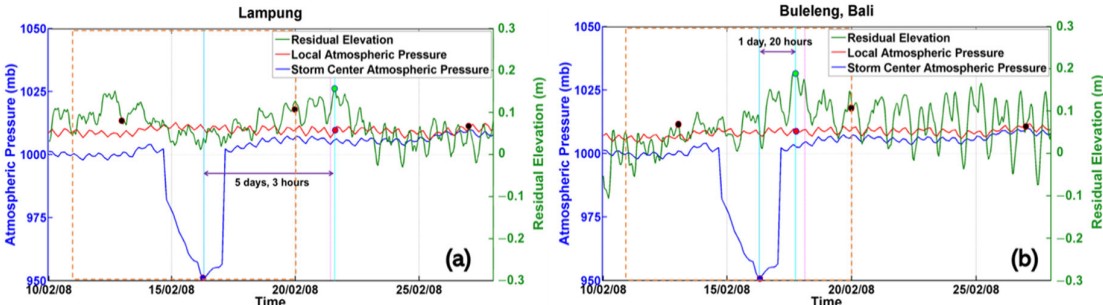

**Figure 3.** Comparison of the residual water levels (non-astronomic tide) at 00:00 UTC on 13, 20, and 28 February 2008. Green line (the simulation results); black dots (the TOPEX/Poseidon Sea Level Anomaly (SLA)); brown dash square (period of TC Nicholas, 11–20 February 2008); blue line (atmospheric pressures at TC Nicholas's center); and red line (local atmospheric pressures). (**a**) At Lampung (marked by Point a in Figure 1); (**b**) at Buleleng (marked by Point b in Figure 1).

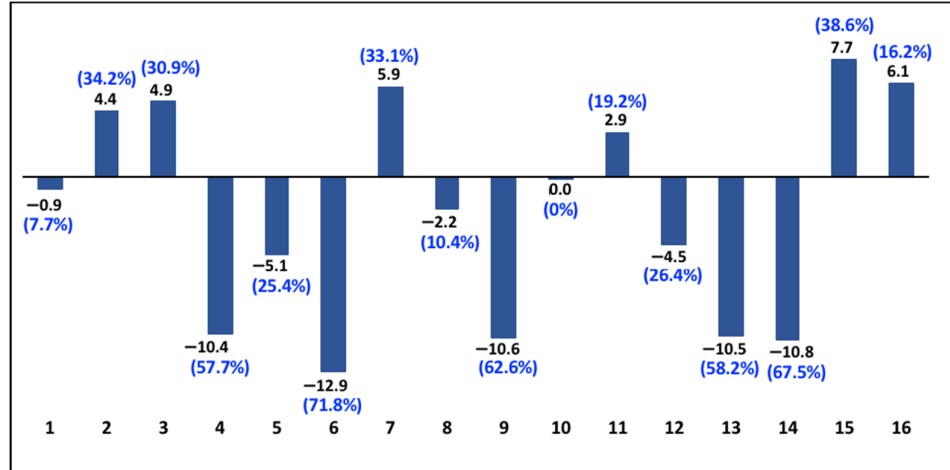

**Figure 4.** Comparison of the maximum values of the non-astronomic tide (residual water level) between the simulated results and those of the TOPEX/Poseidon at 16 sites, which were most vulnerable to the TC Nicholas event (denoted by numbers 1–16; names and locations of the 16 sites can be seen in Table 2 and Figure 6). Black numbers (absolute errors in cm); blue numbers (relative errors in %).

In this study, we attempted to use the limited data availability as optimally as possible for validating meteorological induced sea level variations. Therefore, the performance of the model in simulating them could not be accurately judged because of the lack of observational data. In general, the comparison showed a fair agreement between the simulated maximum values of the residual water level and the TOPEX/Poseidon SLA at the most vulnerable areas affected by TC Nicholas (Figure 6), with mean absolute and relative errors of about 6.2 cm and 35.0%, respectively (Figure 4). On average, Figure 4 shows that the simulated maximum surge residuals overestimated the TOPEX/Poseidon SLA by 6.8 cm (28.7%) at Points 2, 3, 7, 16, and 16, whereas they underestimated the SLA of TOPEX/Poseidon by 7.5 cm (43.1%) at Points 1, 4–6, 8–9, and 12–14. In general, the simulated maximum surge residuals underestimated the TOPEX/Poseidon SLA by 13.5%. The possible source of this underestimated value appears to be that the spatial and temporal resolutions of atmospheric data specified in the model, as well as the TC data, were not sufficiently accurate for the simulation. Moreover, the differences may also be due to the fact that the TOPEX/Poseidon SLA at this spatial/temporal resolution and the estimation of the effect of the bottom friction did not resolve coastal processes well. In this regards, further study is required to address this issue.

### 3.2. Analysis of Increases in Sea Levels Caused by TC Nicholas

Before performing the analysis of maximum values of the increased sea levels due to TC Nicholas, as well as the investigation of the most susceptible areas to the impact of the cyclone, the performance of the model in simulating the increase in sea level at the center of TC Nicholas was quantified (Figure 5). The BOM observed track parameters showed that the minimum atmospheric pressure at the center of TC Nicholas is about 948 mb. Based on the theoretical formulation of IBE, the residual water level around the center of the cyclone is expected to increase by about 65.3 cm as a result of the decrease in atmospheric pressure from its normal value (assuming a normal atmospheric pressure of 1013.3 mb [54]). However, in this study, the simulation results of the increase in sea level at the cyclone center underestimated the theoretical value by about 16.3 cm (24.9%), as shown in Figure 5 (the simulated value of about 49 cm), although we have blended the ERA-Interim ambient atmospheric data with the BOM observed track parameters of TC Nicholas. Better simulation results will be probably obtained if more recent atmospheric data, such as ERA-5 data, as well as other data (e.g., bathymetry) with higher spatial/temporal resolution, and a more adequate estimation of the effect of the bottom friction, are specified in the model. It is also necessary to be aware of the fact that the theoretical formulation of IBE is developed using a large number of

assumptions and approximations. Nevertheless, the present simulation results with a difference of less than 25% from the theoretical value might still be used as an initial assessment to investigate the Indonesian sea areas most prone to the impact of the cyclone.

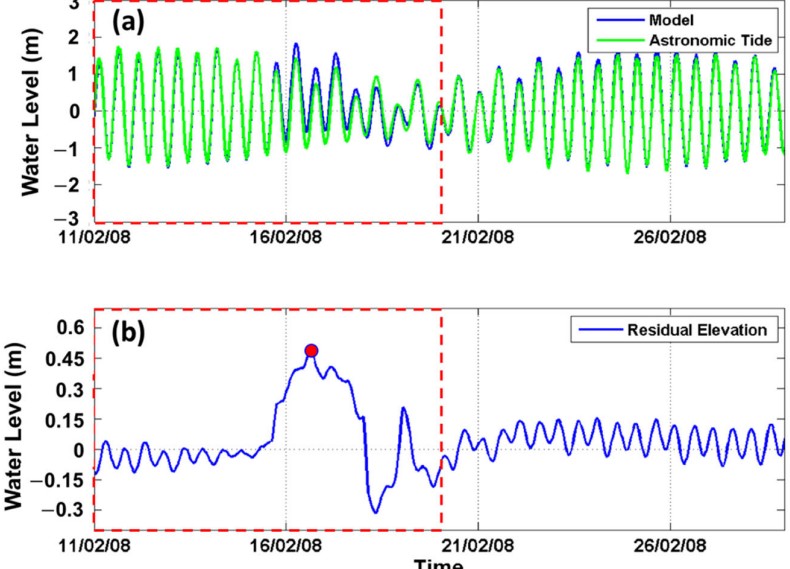

**Figure 5.** The Regional Ocean Modeling System (ROMS) simulation results: (**a**) astronomical (green line) and total sea (blue line) levels and (**b**) residual/non-astronomical sea levels at TC Nicholas's center (marked by Point 15 in Figure 1 and Table 1), with a maximum value of about 49 cm (red dot). Red dash square is the period of TC Nicholas (11–20 February 2008).

3.2.1. Maximum Residual Water Levels and Areas Most Affected by the TC Nicholas

Table 2 and Figure 6 show that there are 16 coastal areas of Indonesia that were most affected by the TC Nicholas, with maximum surge residual values ≥ 10 cm, namely, (a) western coast of Sumatra (Points 1–3); (b) southern coast of Java (Points 4–6); (c) Bali (Points 7–8); (d) Nusa Tenggara (Points 9–10); (e) Belitung Island (Point 11); (f) western coast of Kalimantan (Point 12); (f) South Sulawesi (Point 13); (g) Southeast Sulawesi (Point 14), and (h) western and southern coasts of Papua (Points 15–16). The maximum values of residual waters (non-astronomic tide) at the 16 sites range from 11 to 38 cm, with the two highest residuals occurring at Merauke (38 cm) and Kuta (21 cm), marked by Points 16 and 8 in Figure 6, respectively.

**Table 2.** Areas most affected by the TC Nicholas event, maximum residual water levels, local forcings (wind set-up and inverted barometer effect (IBE)), and remote forcings.

| Points | Locations | Residual Water Levels $(\Delta\eta_{tot})$ (m) | Local Forcings (m) | | | Remote Forcings (m) $\Delta\eta_R = \Delta\eta_{tot} - (\Delta\eta_W + \Delta\eta_{P_a})$ |
|---|---|---|---|---|---|---|
| | | | Wind Set-Up $(\Delta\eta_W)$ | Inverted Barometer Effect $(\Delta\eta_{P_a})$ | Total Local Forcings $(\Delta\eta_W + \Delta\eta_{P_a})$ | |
| 1 | Padang | 0.11 | 0.0001 | −0.0150 | −0.0149 | 0.12 |
| 2 | Bengkulu | 0.13 | 0.0005 | −0.0050 | −0.0045 | 0.13 |
| 3 | Lampung | 0.16 | 0.0018 | 0.0050 | 0.0068 | 0.15 |
| 4 | Garut | 0.18 | 0.0010 | 0.0020 | 0.0030 | 0.18 |
| 5 | Yogyakarta | 0.20 | 0.0017 | 0.0100 | 0.0117 | 0.19 |
| 6 | Malang | 0.18 | 0.0020 | 0.0010 | 0.0030 | 0.18 |
| 7 | Buleleng | 0.18 | 0.0019 | 0.0180 | 0.0199 | 0.16 |
| 8 | Kuta | 0.21 | 0.0008 | 0.0200 | 0.0208 | 0.19 |
| 9 | Koeta | 0.17 | 0.0007 | 0.0220 | 0.0227 | 0.15 |

| 10 | Plampang | 0.18 | 0.0004 | 0.0180 | 0.0184 | 0.16 |
| 11 | Belitung | 0.15 | 0.0100 | 0.0070 | 0.0170 | 0.13 |
| 12 | Jungkat | 0.17 | 0.0057 | −0.0100 | −0.0043 | 0.17 |
| 13 | Makassar | 0.18 | 0.0038 | 0.0100 | 0.0138 | 0.17 |
| 14 | Boepinang | 0.16 | 0.0003 | 0.0100 | 0.0103 | 0.15 |
| 15 | Timika | 0.20 | 0.0030 | 0.0300 | 0.0330 | 0.17 |
| 16 | Merauke | 0.38 | 0.0060 | 0.0280 | 0.0340 | 0.35 |

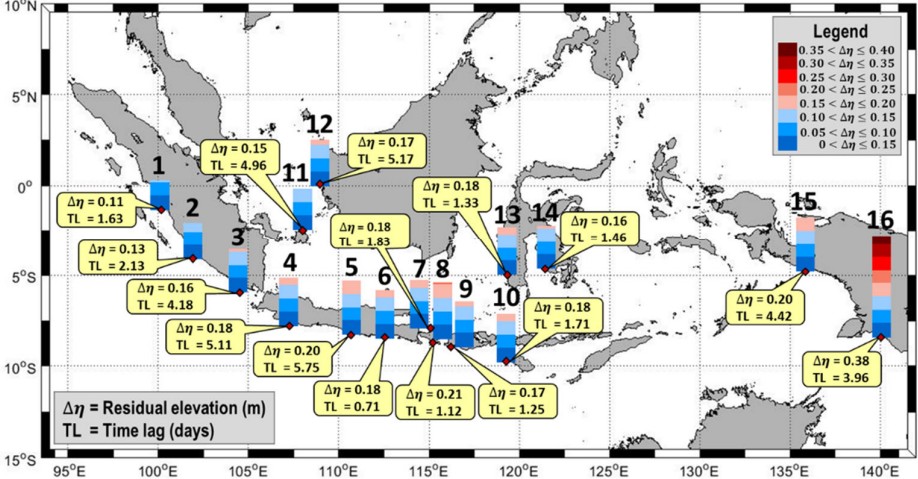

**Figure 6.** The maximum residual sea levels (in cm) at the 16 locations (Points 1–16) that are most susceptible to the effect of TC Nicholas in February 2008 (names of the locations mentioned in Table 2). TL is a time lag between atmospheric pressures at the center of TC Nicholas and residual sea level oscillations at the 16 sites. A positive lag indicates that a decrease in atmospheric pressure at the center of TC Nicholas leads to an increase in residual sea level.

To assess the size of the remote forcings during the TC Nicholas event influencing the increases in sea levels, changes in residual sea levels due to wind set-up and IBE were locally calculated at the 16 sites (Table 2). Their effect (remote forcings) are estimated by subtracting the calculated wind set-up and IBE (local forcings) from the simulated residual sea levels (non-astronomic tide). It is found that the residual sea levels induced by the total local forcings (wind set up and IBE) are about ≤3.4 cm (Table 2). Therefore, it is suggested that the maximum surge residual heights induced by the remote forcings during the TC Nicholas occurrence could reach 12 cm to 35 cm in the Indonesian coastal areas, with a maximum value of about 35 cm at Merauke (Point 16 in Table 2). Moreover, the remote forcings are much more likely to dominate the increased residual sea levels in the Indonesian coastal areas during the cyclone event compared with the local ones. The small effects of the local forcings in rising the sea levels (≤3.4 cm) might be caused not only by both weak local wind speeds (≤15 m/s) and small variations in local atmospheric pressure (≤3 mb), but also by wind directions. During the TC Nicholas event (February 2008), wind fields in the Indonesian seas associated with the northwest monsoon tend to blow parallel to the shoreline of the study areas, so that they generated very small wind-induced set-up (almost 0 cm, as shown in Table 2). However, the wind set-up still slightly occurred in several areas in which the northwest winds blow perpendicular to the coast, such as at Points 11–13, 15, and 16 (Table 2 and Figure 6). For instance, the typical characteristics of the northwest winds, which move towards the coast and generate wind set-up, are shown in Figure 7 at two sites, namely Belitung and Merauke (Points 11 and 16, respectively, in Table 2 and Figure 6). Meanwhile, examples of the small variations in the local atmospheric pressure can be seen in Figure 3.

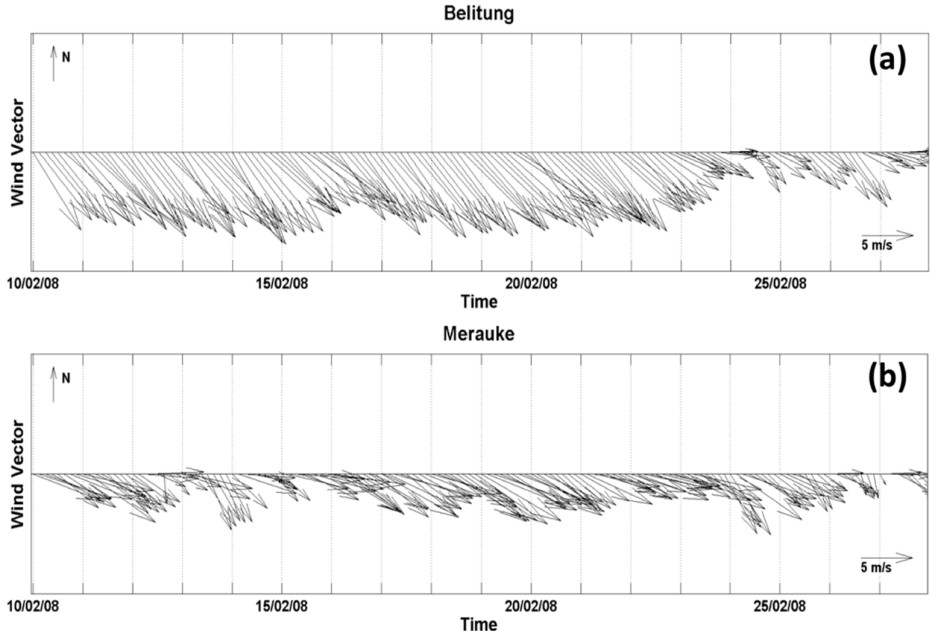

**Figure 7.** Local wind fields at (**a**) Belitung (Point 11 in Figure 6) and (**b**) Merauke (Point 16 in Figure 6).

3.2.2. Relationship of the Residual Water Levels in the Indonesian Waters to the Atmospheric Pressures at TC Nicholas's Center

To confirm the possible effects of the TC Nicholas on the increased residual sea levels in the Indonesian coastal areas, the lagged correlation analysis between them was carried out, as shown in Figure 8, with the 95% significance level of approximately ±0.21. In this regard, we only focused on analyzing negative correlations and positive lags. The negative correlation indicates that any decrease in atmospheric pressure at TC Nicholas's center will be followed by an increase in the residual water level in the Indonesian seas. Meanwhile, the positive lag represents that the variability in a former variable (atmospheric pressure) leads that in the latter variable (residual sea level).

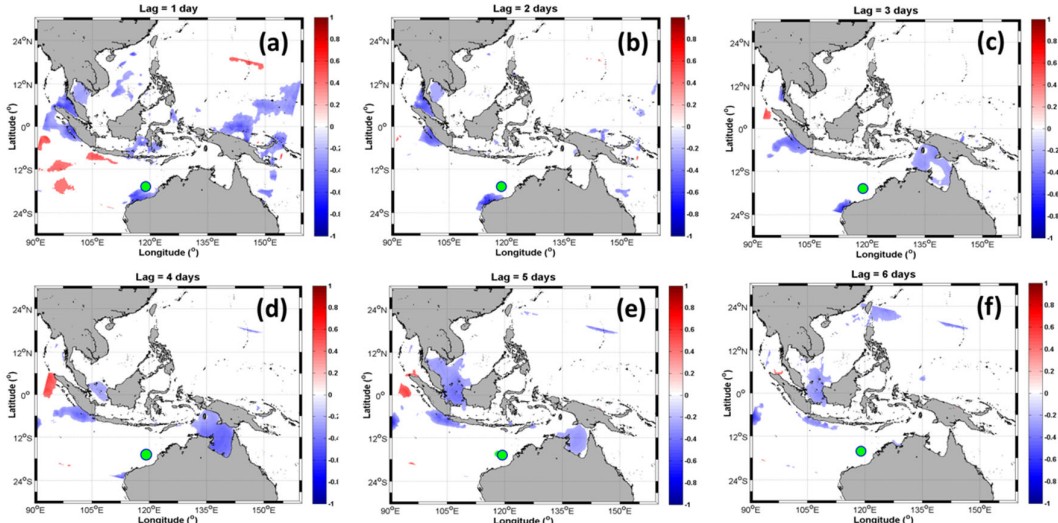

**Figure 8.** Lagged correlation maps between the atmospheric pressures at TC Nicholas's center (green dots) and the residual sea level oscillations in the Indonesian waters for lags of (**a**) 1 day, (**b**) 2 days, (**c**) 3 days, (**d**) 4 days, (**e**) 5 days, and (**f**) 6 days. The 95% significance level is approximately ±0.21. The analysis is only conducted for negative correlations and positive lags. Negative correlations indicate that any decrease in atmospheric pressure at the center of the TC Nicholas will be followed by an

increase in the residual water level in the Indonesian seas. A positive lag indicates that the variability in atmospheric pressure at the cyclone center leads the surge residuals in the Indonesian waters.

Figure 8 shows that there was a strong correlation between the residual sea levels (non-astronomic tides) in the Indonesian coastal areas and the atmospheric pressure at TC Nicholas's center, with the cyclone leading the surge residuals by less than 6 days. This analysis confirmed that the TC Nicholas event had a strong effect on the increased residual water levels in the Indonesian seas. The detailed results of time lags between TC Nicholas and the maximum surge residuals at the most vulnerable coastal areas to the impact of the cyclone (Points 1–16 in Table 2) are shown in Figure 6. This study revealed that the variation of residual sea levels lagged <2 days behind that of TC Nicholas intensity at Point 1 (central part of western coast of Sumatra), Point 6 (southern coast of East Java), Points 7–10 (Bali and Nusa Tenggara waters), and Points 13–14 (South and Southeast Sulawesi). Meanwhile, the surge residuals at Papua (Points 15 and 16) lagged the cyclone by approximately 4.42 days and 3.96 days, respectively. A longer time lag existed at Points 11–12 (Belitung Island and western coast of Kalimantan), which are located farther from the centre of the cyclone compared with the other sites. The surge residuals at the Points 11 and 12 lagged 4.96 days and 5.17 days behind the cyclone, respectively.

Interestingly, it seems that there was a surge residual that propagated eastward as a coastal Kelvin-like wave along the western coast of Sumatra and the southern coast of Java (from Point 1 to Point 5, as can be seen in Figures 6 and 8), with phase speed of approximately 5.52 m/s. The phase speed obtained from this study is slightly stronger than that of [18,19]. Iskandar et al. [18] found that the intraseasonal Kelvin waves (20–90 days), which are attributable to remote wind over the eastern equatorial Indian Ocean (90°E) and the local alongshore wind forcing, propagate with speed ranging from 1.5 to 2.86 m/s, except along the coast of Sumatra (4.91 m/s). Meanwhile, another study by Ningsih et al. [19] revealed the presence of the intraseasonal Kelvin wave (with a spectral peak at 91 days) generated by Madden–Julian oscillation (MJO), which propagates eastward along the area of study with a speed range of 2.56–3.85 m/s.

To further confirm remote wind and atmospheric pressure forcings during TC Nicholas playing an important role in generating residual sea level variability in the Indonesian coastal areas and to identify dominant periods of their variation, we applied a power spectral analysis on both residual sea levels at the observed sites (Points 1–16 in Figure 6) and atmospheric pressures at the center of TC Nicholas. In this paper, as an example, we only show the power spectral analysis at the two sites where the highest residuals took place (Figure 9), namely Kuta (Point 8) and Merauke (Point 16). Figure 9c shows that oscillations with a period of 5.9 and 8.9 days are found in the atmospheric pressures at TC Nicholas's center. These signals also exist in residual sea levels at Kuta with a period of 5.9 days (Figure 9a) and Merauke with a period 8.9 days (Figure 9b), confirming that TC Nicholas significantly contributes to the increase in residual sea levels at both sites.

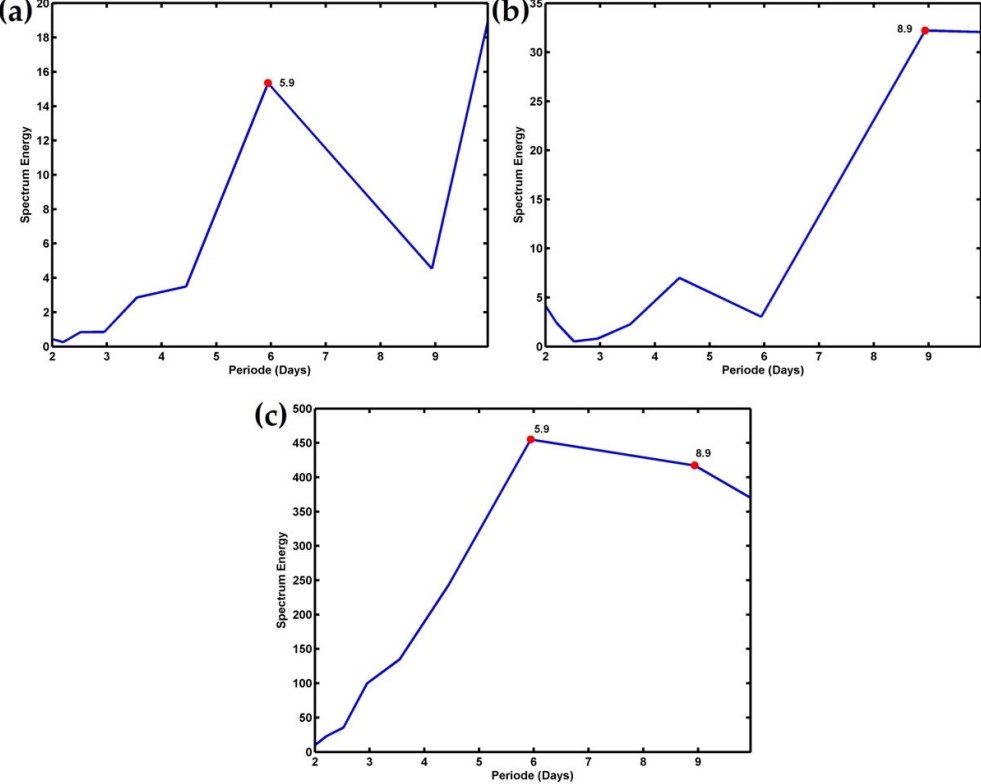

**Figure 9.** Energy spectrum of the residual sea levels at (**a**) Kuta (Point 8 in Figure 6) and (**b**) Merauke (Point 16 in Figure 6). (**c**) Same as in Figure 9a,b, except for the atmospheric pressure at the center of TC Nicholas (Point 15 in Figure 1).

## 4. Conclusions

The effect of TC Nicholas (11–20 February 2008) on the increases in sea levels in the Indonesian waters was investigated by analyzing residual water levels (non-astronomic tide), which were derived by removing the tidal part (astronomic tide) from the ROMS simulated total water levels. It was revealed that the TC Nicholas event had a strong effect on the increased residual water levels in the Indonesian coastal areas. There are 16 coastal areas of Indonesia that were most affected by the cyclone, with maximum surge residual values ranging from 11 to 38 cm and the residuals lagging the cyclone by 0.71–5.75 days. Among the most vulnerable areas to the impact of the cyclone, the two areas most at risk are Kuta (21 cm) and Merauke (38 cm), with the surge residuals lagging by 1.22 days and 3.96 days behind the cyclone, respectively.

In this study, the performance of the model in simulating the surge residuals could not be accurately evaluated because of the lack of observational data. Therefore, we endeavored to use the limited data availability as optimally as possible for verifying the residual sea level variations. The simulated maximum values of the residual water level and the TOPEX/Poseidon SLA at the most vulnerable areas affected by the cyclone are generally in fair agreement, with mean absolute and relative errors of about 6.2 cm and 35.0%, respectively. In addition, the simulated maximum surge residuals at the areas most affected by the cyclone underestimated the TOPEX/Poseidon SLA by 13.5% and the simulated value of the increase in sea level at the cyclone center also underestimated the theoretical value by about 24.9%. Accordingly, it is necessary to carry out future research, which includes detailing the forcing mechanisms, as well as using both higher spatial and temporal resolutions of data input and model grid and specifying more adequate estimation of the effect of the bottom friction to obtain a better simulation result of this important and interesting topic. These kinds of research works are currently in progress as an extension of the present study.

Though the performance of the present model needs to be improved, the present simulation results might still be used as an initial estimation to study the most susceptible Indonesian coastal regions to the impact of the cyclone. A better understanding of the cyclone effect on the residual sea levels in Indonesian waters could be valuable for a disaster management plan, especially in the coastal areas, and primarily to reduce the risk of flooding due to the tropical cyclone.

**Author Contributions:** N.S.N.: conceptualization, methodology, formal analysis, investigation, resources, writing—review and editing, supervision, project administration, funding acquisition; F.H.: methodology, software, formal analysis, writing—original draft preparation, writing—review and editing, visualization; T.S.T.: methodology, software, validation, data curation, visualization; L.F.Y.: software, validation, data curation, editing, visualization; M.A.A.: Supervision in utilization of hydrodynamic modelling software. All authors have read and agreed to the published version of the manuscript.

**Funding:** Parts of this research were funded by the Indonesian Ministry of Research and Technology/National Research and Innovation Agency (Kemenristek/BRIN) under the research program of Penelitian Dasar (PD) 2020 (Research Grand Contract No. 2/AMD/E1/KP.PTNBH/2020).

**Acknowledgments:** We gratefully acknowledge the support of Institut Teknologi Bandung (ITB) and the Kemenristek/BRIN.

**Conflicts of Interest:** The authors declare that they have no conflict of interest.

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
