# Peer review of "The Effect of Tropical Cyclone Nicholas (11–20 February 2008) on Sea Level Anomalies in Indonesian Waters"

_jmse, doi:10.3390/jmse8110948_

Round 1

Reviewer 1 Report

This study reports effect of Tropical Cyclone (TC) Nicholas on the rise of sea level in Indonesian waters using the Regional Ocean Modeling System (ROMS). I have reviewed this paper before and I suggested several changes at the time, mainly in the introduction and discussion sections. Overall, this version has several improvements over the previous version following most of my comments and suggestions. Despite the improvements made to the introduction, I think it remains a description of the study area. Therefore, I suggest changing the introduction again in order to expose the problem under study more broadly, presenting other studies carried out in other regions. 

Reviewer 2 Report

The authors have made significant changes to the manuscript since the first review and have addressed all of my initial comments. As a result, the MS is much improved and I believe is nearly ready for publication in JMSE, pending addressing some final minor comments below:

  1. Title – given the confusion acknowledged between “sea level rise” and “sea level fluctuations”, I would suggest modifying the title to “…on sea level anomalies in Indonesian Waters” rather than “… on increases in sea levels in Indonesian Waters”
  2. Abstract - explains results and context clearly, but requires an extra sentence at the end to clarify the relevance and impact of these findings – you have done this in the Conclusions, which is good.
  3. Bathymetry resolution – you have provided the scale of the nautical chart (1:20,000) – it would be more useful to quote an average sounding spacing so that the data point density can be compared to the model computational grid resolution.
  4. Parametric representation of TC wind field – you say you blended ERA-Interim winds with a parametric TC wind field ; please state which parametric model you used to generate this wind field and blend with ambient winds

Author Response

This manuscript is a resubmission of an earlier submission. The following is a list of the peer review reports and author responses from that submission.

Round 1

Reviewer 1 Report

The manuscript “The effects of Tropical Cyclone Nicholas … on the Rise of Sea Level in Indonesian Waters” describes a modelling study looking at the effects of non-astronomic water levels around Indonesia from a Tropical Cyclone event. The modelling work appears to be generally well executed, and the paper well written with the results having potential for regional interest. However, I have some serious concerns around broader relevance, scientific depth of analysis and methodology in the present manuscript. While I believe these can all be addressed by the authors, I recommend for this work to be re-submitted for review when these changes have been made.

General Comments

  1. Why did you only focus on a single TC event? Why did you choose a relatively old (2008) and weak (Cat 1) event for this analysis? Either, the depth of analysis related to this single event needs to be improved, and the results presented in the context of the wider cyclone climatology for the area, or other notable events could be included in the analysis (i.e. perhaps those of greater strength).
  2. You refer to “sea level rise” throughout the paper but this term is generally reserved for long-term rise in water levels associated with climate change. You are really looking at sea level fluctuations, or deviations. Also, you refer to long waves and short waves, by which I presume you mean infragravity and gravity surface waves? However, this appears to be a hydrodynamic and not a wave modelling study. It seems you are looking at surge residuals – the difference between the astronomic and non-astronomic component of total water levels. The terminology throughout needs to be changed to reflect this.
  3. The conclusions are very short, as is the reference list. The brevity of both, and the abstract, all reflect the general lack of discussion around the relevance, context and impact of this work. At present, it reads like a report. What is the “so-what?” of the work? Why does this work matter, why is it important? Some of the conclusions are very obvious i.e. "the farther the location from the centre of the cyclone, the smaller the effect on sea level rise". There needs to be a greater depth of analysis for this to be publishable.

Focussed Comments

  1. Line 50 – “a significant increase of up to 1028.31 % in wave height” – a strange way to express this; better to give real values compared to mean annual wave height values. I think this is also reflective of the confusion between waves and surge – why are wave heights relevant here when the study is not focused on surface ocean waves?
  2. Line 56 – Should be “detailed”, not “detail”
  3. Figure 1 – it would be useful to include the TC track on this plot
  4. Line 93 - What resolution was the HOCIN bathymetry at? Because it seems your grid is more highly resolved than the base bathymetry? Is this the case? There should be some comparison between bathymetry / computational grid resolutions. Was any sensitivity testing doe to optimise the model configuration?
  5. Line 103 - more information is needed about how the TC wind field was accounted for - ERA-Interim at this spatial/temporal resolution will not capture the intensity of the TC correctly. Did you blend these 'ambient' winds with a parametric representation of the TC wind field from the observed track parameters? Why did you choose ERA-Interim and not the more recent ERA5?
  6. Line 122 – How were sea levels harmonically analysed? What package and method was used?
  7. Table 1 – give units of RMSE
  8. Line 149 – It appears you are comparing the astronomic component of the modelled water levels (i.e. tide), with tidal predictions? By doing so, are you not just comparing TOPEX tides (which are the boundary tides used in your model) with BIG tidal predictions? It is unclear to me why you are not comparing total water levels from the model to observations? Or, better still, run the model twice; one with tide only, then tide plus winds, subtract one from the other to get the modelled surge residual, and compare this to the observed surge residual (i.e. observations with the harmonic component removed)?
  9. Line 161 – the difference between observed residual and inverted barometric value may be related to the ambient atmospheric pressure value you used in the IB calculation. For example, an atmospheric pressure of 1000 mb is quite low – usually 1030 is used.
  10. Line 170 - what do you mean by “remote forcing”? From Table 2, remote forcing looks to be the difference between [the observed surge residual at the gauge sites] – [the calculated IBC and wind set up.]
  11. Figure 4 - this is a nice idea, but bear in mind that correlation does not mean causation - i.e. you may find areas of correlation at longer lag times than 5 days (did you?) but this doesn't mean there is a physical connection between the two. Can you confirm you are only showing R values > 95 % confidence level? What R value is the 95 % confidence level? In the caption you say the 99 % significant level is -0.21. Can you explain why there is no positive correlation in Lag = 0 directly to the west of the green dot (i.e. the patch of red only appears further west)
  12. Figure 5 - what do these residual elevations refer to? Are they modelled data points with the tidal component removed?

Reviewer 2 Report

General comment

This study reports effect of Tropical Cyclone (TC) Nicholas on the rise of sea level in Indonesian waters using the Regional Ocean Modeling System (ROMS). This is a relevant topic in the actual scientific context, however the manuscript has limitations that compromise its publication in JMSE. First, there is no introduction to the topic under study. The introduction content is mainly a description of the study area. Second, the model was verified for tides, however its performance in simulating meteorological induced sea level rise is unknown. This is essential, since the model was applied to study the meteorological induced sea level rise. Third, there is no a discussion of the results. There is a section named Results and Discussion that describes the results and did not discuss the results. The entire manuscript presents 11 references, which is clearly insufficient for a scientific research paper.

Specific comments

Introduction

The introduction presented is a description of the study area. Your introduction presents 7 citations, all of them are local works. Please improve this section following the journal instructions:

https://www.mdpi.com/journal/jmse/instructions

“The introduction should briefly place the study in a broad context and highlight why it is important. It should define the purpose of the work and its significance, including specific hypotheses being tested. The current state of the research field should be reviewed carefully and key publications cited. Please highlight controversial and diverging hypotheses when necessary. Finally, briefly mention the main aim of the work and highlight the main conclusions. Keep the introduction comprehensible to scientists working outside the topic of the paper.”

I suggest adding a new subsection entitled Study Area at the beginning of Materials and Methods section. Almost all the contents of the present in the introduction should be add into the new subsection Study Area.

Materials and Methods

Please include a summary of the entire methodology at the end of the introduction or at the beginning of the methods to guide the reader through the procedures. This information clarifies the work and facilitates the reading and understanding of the work.

[74] Please correct this reference. Pawlowicz [9] did not develop the numerical model ROMS.

[122-123] What is the method used to separate astronomical and non-astronomical components? Please detail, and include the necessary references.

[110-114] In this work you validate the model performance in simulating tidal levels, however the model was used to simulate sea level variations induced by meteorological effects. How do you know the model performance in simulating meteorological induced sea level variations? It is fundamental to include the validation of non-astronomical tide during past Tropical Cyclone events.

Results and discussion

This section should be named results, since discussion pieces are almost inexistent.

I suggest adding a discussion section following the jmse recommendations:

https://www.mdpi.com/journal/jmse/instructions

“Discussion: Authors should discuss the results and how they can be interpreted in perspective of previous studies and of the working hypotheses. The findings and their implications should be discussed in the broadest context possible and limitations of the work highlighted. Future research directions may also be mentioned. This section may be combined with Results.”

[164-165] However your model can underestimate or overestimate peaks. You need to include a validation of meteorological induced sea levels during past Tropical Cyclones.

[170-173] This is concordant with previous studies? Explain why local winds and local atmospheric pressure have very little effect compared to atmospheric pressure at the center of the cyclone? Complete explanations with appropriate references.